# ADVERSARIAL READING NETWORKS FOR MACHINE COMPREHENSION

## ABSTRACT

Machine reading has recently shown remarkable progress thanks to differentiable reasoning models. In this context, End-to-End trainable Memory Networks (`MemN2N`) have demonstrated promising performance on simple natural language based reasoning tasks such as factual reasoning and basic deduction. However, the task of machine comprehension is currently bounded to a supervised setting and available question answering dataset. In this paper we explore the paradigm of adversarial learning and self-play for the task of machine reading comprehension. Inspired by the successful propositions in the domain of game learning, we present a novel approach of training for this task that is based on the definition of a coupled attention-based memory model. On one hand, a *reader network* is in charge of finding answers regarding a passage of text and a question. On the other hand, a *narrator network* is in charge of obfuscating spans of text in order to minimize the probability of success of the reader. We experimented the model on several question-answering corpora. The proposed learning paradigm and associated models present encouraging results.

## 1 INTRODUCTION

Automatic comprehension of text is one of the main goals of natural language processing. While the ability of a machine to understand text can be assessed in many different ways, several benchmark datasets have recently been created to focus on answering questions as a way to evaluate machine comprehension (Richardson et al., 2013); (Hermann et al., 2015); (Hill et al., 2015a); (Weston et al., 2015); (Rajpurkar et al., 2016); (Nguyen et al., 2016). In this setup, the machine is presented with a piece of text such as a news article or a story. Then, the machine is expected to answer one or multiple questions related to the text. The task is linked to several important incomes. First, it provides tools that will shortly help users with efficient access to large amounts of information. Also, it acts as an important proxy task to assess model of natural language understanding and reasoning. In this context, numerous large-scale machine comprehension/QA datasets (Hermann et al., 2015); (Rajpurkar et al., 2016); (Trischler et al., 2016a); (Nguyen et al., 2016) have been recently released and have contributed to significant advancement. From a model perspective, neural models are now approaching human parity on some of these benchmarks and a large corpus of novel and promising research has been produced in the domain of attention, memory and parametric model with so-called reasoning capabilities. However, the field is currently bounded to the paradigm of supervised learning and strictly linked to the current annotated dataset. As a counterpart, an increasing research activity has been dedicated since the 90's to self-play and adversariality to overcome this boundary and allow a model to exploit its own decision to improve itself. Two famous examples are related to policy learning in games. Indeed, TD-Gammon (Tesauro, 1995) was a neural network controller for backgammon which achieved near top player performance using self-play as learning paradigm. More recently, DeepMind AlphaGo uses the same paradigm to win against the current world best human go player. The major advantage of such setting is to partially release the learning procedure to the limit of an available dataset. The dual models learn and improve their performance by acting one against the other as so-called sparing patterns.

In this paper, we adapt this paradigm to the domain of machine reading. On the first hand, a **reader network** is trained to learn to answer question regarding a passage of text. On the other hand, a **narrator network** learns to obfuscate words of a given passage in order to minimize the probability of successfull answering of the reader model. We developed a sequential learning protocol in order

to gradually improved the quality of the models. This paradigm separates itself from the current research direction of joint question and answer learning from text as proposed on Wang et al. (2017). Indeed, in comparison to question generation as regularizer of a reader model that sounds promising, we believe adversarial training unleashs from the constraint of strict and bounded supervision and brings robustness to the answering model.

Our contributions can be summarized as follows: (1) We propose a new learning paradigm for machine comprehension based on adversarial training. (2) We show this methodology allows to overcome the boundaries of strict supervision and provides robustness to noise in question-answering settings through a set of experiments in several machine reading corpora and (3) visualizations of the models reveals some useful insights of the attention mechanism for reasoning the questions and extracting meaning passage of a text given a question.

**Roadmap:** In Section 2, we formalize our adversarial learning protocol. Also, the reader and narrator networks are presented. In Section 3 the corpora used for evaluation are detailed. Section 4 presents our current experimental results. Section 5 details several vizualizations of the decisions and attention values computed by the coupled models. Finally, Section 6 reviews the state-of-the-art of machine reading comprehension, Memory Network models, the paradigm of self-play and its links to adversarial learning.

## 2 ADVERSARIAL READING NETWORKS

Several studies have recently challenged deep machine reading models with adversarial examples as Miyato et al. (2016) and Jia & Liang (2017). This kind of approach is well known in computer vision (Goodfellow et al., 2014) but seems to also affects natural language processing. More precisely, Jia & Liang (2017) demonstrates that a large majority of the recent state of the art deep machine reading models suffers from a lack of robustness regarding adversarial examples because of their so-called oversensibility. Indeed average accuracies were decreased by half when these models were tested on corrupted data, i.e a document with an additional sentence at the end which normally does not affect the answer. The model we propose is built to use this adversariality as an adaptive dropout by challenging the reader with more and more difficult tasks during the learning. Indeed, we extend the concept of *asymmetric self-play* to train a model that we called the *narrator* during an adversarial game with a *reader*. The narrator is acquiring knowledge about the reader behaviour during the training and it generates harder adversarial examples. Beyond increasing artificially the size of the available dataset, this adaptive behaviour of the narrator prevents catastrophic forgetting phenomena from the reader. In this section, we explain the protocol of adversarial training we developed for robust machine comprehension. Then, we describe the reader and narrator models used.

### 2.1 MAIN LEARNING PROTOCOL

The overall framework is a turn-based question answering game described in Figure 1. At the beginning of each round, the narrator obfuscates one word for each document sampled from the training corpus. We fix the ratio of corrupted data / clear data to a ratio $\lambda \in \mathbb{R}^{[0,1]}$ of the dataset. Indeed, a too low percentage of corrupted data might not have any effect on the training and a too high one will prevent the reader of learning well. Then, the reader is trained on a subset of this obfuscated corpus and tested on the remaining subset. Note that both train and test sets contain corrupted data. Finally the narrator gets back a set of rewards regarding the reader performances on the obfuscated stories. Given a tuple $(d, d_{\text{obf}}, q)$ where $d$ is the original document, $d_{\text{obf}}$ the document with an obfuscated word proposed by the narrator and $q$ the associated question, the reward $r$ given to the narrator is defined as follow:

$$r = \begin{cases} 1 & \text{if the reader answer well on } d \text{ and fail on } d_{\text{obf}} \\ 0 & \text{otherwise} \end{cases}$$

The reward given to the narrator is a direct measurement of the impact of the obfuscation on the reader performance. All the previously collected rewards are stored and used for experience replay throughout the turns. After each learning turn, all the parameters of the narrator are reinitialized and retrained on all the recorded rewards. Throughout the turns, the narrator accumulates information

about the reader behaviour and proposes more challenging tasks as the game is playing. Each narrator's dataset is choosen to maximizes its expected rewards for 80% of the stories and randomly obfuscates a word in the remaining 20% in order to ensure exploration. Finally, the reader keeps improving through the turn and any catastrophic forgetting is compensated at the next turn of the narrator by especially focusing on these flaws.

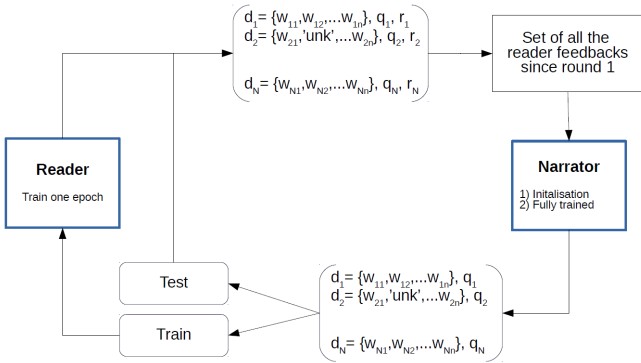

Figure 1: Adversarial learning protocol

---

**Algorithm 1** Pseudo-code of the adversarial training

---

Split dataset into 3 pieces $(A)$ train (80%), $(B)$ valid (10%) and $(C)$ test (10%)
Create $D$ an empty dataset
epoch = 0
**while** epoch < NB_MAX_EPOCHS **do**
    Split $A$ into $A1$ (80%) and $A2$ (20%)
    **if** epoch = 0 **then**
        Randomly corrupt 20% of $A1$ and 100% of $A2$
    **else**
        Reinitialize all the parameters of the narrator
        Train the narrator on $D$
        The narrator corrupts 20% of $A1$ and 100% of $A2$
    **end if**
    Train one epoch of the reader on $A1$
    Let $A2\_clear$ be the dataset that contains the same data as in $A2$ but without corruption
    Test the reader on $A2$ and on $A2\_clear$
    **for all** $(d \in A2, d\_clear \in A2\_clear)$ **do**
        Let $r$ be the reward given to the narrator
        **if** The reader succeed on $d\_clear$ and fails on $d$ **then**
            $D \leftarrow \{D \cup (d, r = 1)\}$
        **else if** The reader succeed on $d\_clear$ and succeed on $d$ **then**
            $D \leftarrow \{D \cup (d, r = 0)\}$
        **end if**
    **end for**
    Test the reader on $B$ and see if it should early stop or not
    epoch $\leftarrow$ epoch + 1
**end while**
Test the reader on $C$ and report the results

---

Finally let $\widehat{a}$ be the predicted distribution and $a$ the the ground-truth.
Categorical cross entropy $L_{\text{Narrator}} = -\sum_{i=1}^{N} \sum_{j=1}^{v} a_{ij} \log(\widehat{a}_{ij})$, is the loss function for the reader network as the model decision is a distribution over a vocabulary. Then, binary cross entropy $L_{\text{Reader}} = -\sum_{i=1}^{N} [a_i \log(\widehat{a}_i) + (1 - a_i) \log(1 - \widehat{a}_i)]$ is used as loss function for the narrator network.

## 2.2 BASELINE PROTOCOL

As a reference protocol, one word is obfuscated in several stories of the dataset using a uniform sampling strategy. This is a naive variation of the first protocol where the narrator doesn't learn from the reader feedbacks. In fact, this protocol is similar to a dropout regularization that allows to avoid overfitting the training set. However without the narrator learning of the first protocol, we lost the adaptive dropout and all the curriculum learning notions of easier and harder inputs. In practice this simple adversarial protocol improves the robustness of the results compared to a standard learning protocol. This learning protocol have strong similarities with the one proposed by Maaten et al. (2013).

## 2.3 READER NETWORK

We use a Gated End-to-End Memory Network, `GMemN2N`, as reader which was first introduced by Perez & Liu (2016). This architecture is based on two different memory cells and an output prediction. An input memory representation $\{m_i\}$ and an output representation $\{c_i\}$ are used to store embedding representations of inputs. Suppose that an input of the model is a tuple $(d, q)$ where $d$ is a document, i.e. a set of sentences $\{s_i\}$ and $q$ a query about $d$, the entire set of sentences is converted into input memory vectors $m_i = A\Phi(s_i)$ and output memory vectors $c_i = C\Phi(s_i)$ by using two embedding matrix $A$ and $C$. The question $q$ is also embedded using a third matrix $B$, $u = B\Psi(q)$ of the same dimension as $A$ and $C$. where $\Phi$ and $\Psi$ are respectively the sentence embedding function and the question embedding function described in the next paragraph. The input memory is used to compute the relevance of each sentence in its context regarding the question, by computing the inner product of the input memory sentence representation with the query. Then a softmax is used to compute the probability distribution. The response $o = \sum_i p_i c_i$ from the output memory is the sum of the output memory vectors $\{c_i\}$ weighted with the sentence relevances calculated before $p_i = \text{softmax}(\boldsymbol{u}^T m_i)$, where $\text{softmax}(a_i) = e^{a_i} / \sum_{j \in [1,n]} e^{a_j}$. A gated mechanism is used when we updated the value of the controller $u$:

$$T^k(u^k) = \sigma(W_T^k u^k + b_T^k)\mathbf{u^{k+1}} = o^k \odot T^k(u^k) + u^k \odot (1 - T^k(u^k)) \tag{1}$$

Finally, assuming we use a model with $K$ hops of memory, the final prediction is $\hat{a} = \text{softmax}(W(o^K + u^K))$ where $W$ is a matrix of size $d \times v$ and $v$ is the number of candidate answers. In this model, we do not use the adjacent or layer-wise weight tying scheme and all the matrix $A^k$ and $B^k$ of the multiple hops are different.

**Text and question representations:** To build the sentence representations, we use a 1-dimensional Convolutional Neural Network (CNN) with a list of filter sizes over all the sentences as proposed in Kim (2014). Let $[s_1, \ldots, s_N]$ be the vectorial representation of a document with $N$ sentences where $s_i = [w_{i,1}, w_{i,2}, \ldots, w_{i,n}]$ is the $i - th$ sentence which contains $n$ words. Given a convolutional filter $F \in \mathbb{R}^{h*d}$ where $h$ is the width of the convolutional window, i.e the number words it overlaps, the convolutional layer produces:

$$c_{i,j} = f(F \odot [Ew_{i,j}, \ldots, Ew_{i,j+h}]), \forall j \in [1, n-j]$$

where $\odot$ is the elementwise multiplication, $f$ a rectified linear unit (ReLU), $b$ a bias term and $E$ the embedding matrix of size $d * V$ where $V$ is the vocabulary size and $d$ the word embedding size. Then, a max pooling operator is applied to this vector to extract features. Given a filter $F$, after a convolutional operation and a max pooling operation, we obtain a feature $\hat{c}_i = \max_j(c_{i,j})$ from the $i - th$ sentence of the text. Multiple filters with varying sizes are used. Assume that our model uses $N_s$ different filter sizes and $N_f$ for each size, we are able to extract $N_s \times N_f$ features for one sentence. The final representation of the sentence $\Phi(s_i) = [\hat{c}_{i F_1}, \hat{c}_{i F_2}, \ldots, \hat{c}_{i F_{N_s * N_f}}]$ is the concatenation of the extracted features from all the filters.

## 2.4 NARRATOR NETWORK

The objective of this model is to predict the probability of the reader to successfully respond to a question given a document with an obfuscated word. This information will be use by the narrator to determine the position of the obfuscated word in the document which maximizes the probability

of the reader to fail its task. We use a `GMemN2N` similarly to the reader. However, on the last layer a sigmoid function is used to predict the probability of the reader to fail on this input: $\widehat{a} = \sigma(W(o^K + u^K))$ where $\sigma = \frac{1}{1+e^{-x}}$ and $\widehat{a} \in [0, 1]$ is the predicted probability of failure of the reader and $W$ a matrix of size $d \times 1$.

An input of the reader is a tuple $(d_{\text{obf}}, q)$ where $d_{\text{obf}}$ is a document with an obfuscated word. To obfuscate a word, we replace it by the word *unk* for *unknown*. The output of the narrator is a real number $r \in \mathbb{R}^{[0,1]}$ which is the expected probability of the reader to fail on the question. The objective of the narrator is to select the stories which maximize this reward. Finally, we use the same text passage and query representation than for the reader, based on a CNN with different filter sizes for the document and the two last hidden states of a bidirectional Gated Rectified Unit (GRU) recurrent network for the question encoding. Both models are fully-differentiable.

## 3 Datasets and Data Preprocessing

**Cambridge Dialogs**: the transactional dialog corpus proposed by Wen et al. (2016) has been produced by a crowdsourced version of the Wizard-of-Oz paradigm. It was originally designed for dialog state tracking but Perez (2016)) have shown that this task could also be considered as a reading task. In such setting, the informable slots provided as metadata to each dialog were used to produce questions for a dialog comprehension task. The dataset deals with an agent assisting a user to find a restaurant in Cambridge, UK. To propose the best matching restaurant the system needs to extract 3 constraints which correspond to the informable slots in the dialog state tracking task: *Food, Pricerange, Area*. Given a dialog between an agent and a user, this informable slots become questions for the model we propose. The dataset contains 680 different dialogs about 99 different restaurants. We preprocess the dataset to transform it into a question answering dataset by using the three informable slot types as questions about a given dialog. After this preprocessing operation, we end up with our question answering formatted dataset which contains 1352 possible answers.

**TripAdvisor aspect-based sentiment analysis**: the dataset contains hotel reviews from the TripAdvisor website (Wang et al., 2010). This dataset contains a total of 235K detailed reviews about 1850 hotels. Each review is associated to an overall rating, between 0 and 5 stars. Furthermore, 7 aspects: *value, room, location, cleanliness, checkin/front desk, service,* and *business service* are available. We transform the dataset into a question answering task over a given review. Concretely, for each review a question is an aspect and we use the number of stars as answer. This kind of machine reading approach to sentiment analysis was previously proposed in Tang et al. (2016).

**Children's Book Test (CBT)**: the dataset is built from freely available books (Hill et al., 2015b) thanks to Project Gutenberg[1]. The training data consists of tuples $(S, q, C, a)$ where $S$ is the *context* composed by 20 consecutive sentences from the book, $q$ is the *query*, $C$ a set of 10 *candidate answers* and $a$ the *answer*. The query $q$ is the $21^{st}$ sentence, i.e the sentence that directly follows the 20 sentences of the *context* and where one word is removed and replaced with a missing word symbol. Questions are grouped into 4 distinct categories depending of the type of the removed word: Named Entities (NE), (Common) Nouns (CN), Verbs (V) and Prepositions (P). The training contains $669, 343$ inputs (context+query) and we evaluated our models on the provided test set which contains $10, 000$ inputs, $2, 500$ per category.

## 4 Experiments

### 4.1 Training Details

10% of the dataset was randomly held-out to create a test set. We split the dataset before all the training operations and each of the protocol we propose was tested on the same test dataset. For the training phase, we split the training dataset to extract a validation set to perform early stopping. We use Adam optimizer (Kingma & Ba, 2014) with a starting learning rate at 0.0005. We set the dropout to 0.9 which means that during training, 10%, randomly selected, of the parameters are not used during the forward pass and not updated during the backward propagation of error. We also added the gated memory mecanism of Perez & Liu (2016) that dynamically regulates the access

---

[1]`https://www.gutenberg.org`

to the memory blocks. This mechanism had a very positive effect on the overall performances of our models. All weights are initialized randomly from a Gaussian distribution with zero mean and $\sigma = 0.1$. Moreover, we penalize the loss with the sum of the $L_2$ of the parameters of the models.

We set the batch size to 16 inputs and we use embedding word of size 300. We initialize all the embedding matrix with pre-trained `GloVe` word vectors (Pennington et al., 2014) and we randomly initialize the words of our document that are not in the `GloVe` model. It seems that for our experiments CNN encoding doesn't improve only the overall accuracy of the model compared to LSTM but also the stability by decreasing the variance of the results. So in practice we use 128 filters of size 2, 3, 5 and 8 so a total of 512 filters for the one dimensional convolutional layer.

We repeat each training 10 times for the two first datasets and report maximum and average accuracy on the test set. The maximum is the score on the test set of the best of the 10 trained models based on the validation set. During the adversarial learning, the dataset contains 70% of clear dialogs and 30% of corrupted dialogs, $\lambda = 0.3$. Inside these corrupted data, 20% are randomly obfuscated by the narrator in order to make it learn from exploration and the narrator maximizes his reward for the remaining 80%. Eventually to fit with the format of the dataset, we slightly modified the output layer of our reader for the CBT task. Instead of projecting on a set of candidate answers the last layer of the reader makes a projection on the entire vocabulary $\hat{a} = \sigma(M \odot W(o^K + u^K))$ where $W$ is a matrix of size $V * d$ with $V$ the vocabulary size, $\odot$ the elementwise product and $M$ the mask vector of size $V$ containing 1 if the corresponding word is proposed in the candidate answers 0 otherwise.

## 4.2 RESULTS

| hops | Log Reg | ASR | GMemN2N | | | uniform GMemN2N | | | adversarial GMemN2N | | |
|---|---|---|---|---|---|---|---|---|---|---|---|
| | | | 4 | 5 | 6 | 4 | 5 | 6 | 4 | 5 | 6 |
| Max | 58.4 | 40.8 | 82.1 | 85.8 | 80.6 | 85.1 | 85.8 | 82.8 | 82.8 | 79.8 | **88.1** |
| Mean | 58.2 | 39.5 | 76.9 | 74.8 | 74.2 | 77.4 | 77.7 | 74.9 | **79.8** | 77.8 | 79.6 |

Table 1: Average and maximum accuracy (%) on the Cambridge dataset on 10 replications of our `GMemN2N`, uniform `GMemN2N` and adversarial `GMemN2N`

| hops | Log Reg | ASR | GMemN2N | | | uniform GMemN2N | | | adversarial GMemN2N | | |
|---|---|---|---|---|---|---|---|---|---|---|---|
| | | | 4 | 5 | 6 | 4 | 5 | 6 | 4 | 5 | 6 |
| Max | 59.4 | 45.2 | 62.3 | 62.4 | 60.5 | 63.1 | 61.4 | 63.1 | **64.6** | 63.5 | 62.3 |
| Mean | 59.0 | 42.3 | 60.8 | 60.6 | 58.5 | 62.3 | 60.3 | 59.6 | **62.8** | 61.2 | **60.8** |

Table 2: Average and maximum accuracy (%) on the TripAdvisor dataset on 10 replications of our `GMemN2N`, uniform `GMemN2N` and adversarial `GMemN2N`

| Task | Log Reg | | | | ASR | | | |
|---|---|---|---|---|---|---|---|---|
| | P | V | NE | CN | P | V | NE | CN |
| Max | 56.3 | 37.1 | 26.5 | 25.6 | 24.7 | 32.7 | 22.1 | 18.3 |

| Task | GMemN2N | | | | uniform GMemN2N | | | | adversarial GMemN2N | | | |
|---|---|---|---|---|---|---|---|---|---|---|---|---|
| | P | V | NE | CN | P | V | NE | CN | P | V | NE | CN |
| Max | 56.0 | 58.5 | 31.9 | 39.0 | 58.1 | 53.6 | 31.6 | 34.0 | **71.1** | **60.4** | **35.3** | **39.4** |

Table 3: Accuracy (%) on the CBT dataset for our `GMemN2N`, uniform `GMemN2N` and adversarial `GMemN2N`

Performance results on the Cambridge dataset and TripAdvisor are displayed in table 2. We present the results of our implementation of a standard `GMemN2N`, a *uniform* `GMemN2N` which is the reader trained with the baseline protocol 2.2 and the `GMemN2N` trained in the adversarial protocol 2.1 against the narrator. Each of the experiment was run 10 times and we report in this table the maximum score on the test (based on validation set) and the average score. The precise number of hops needed to achieve the best performance with such models is not obvious so we are presenting all the results for reader and narrator between 4 and 6 hops.

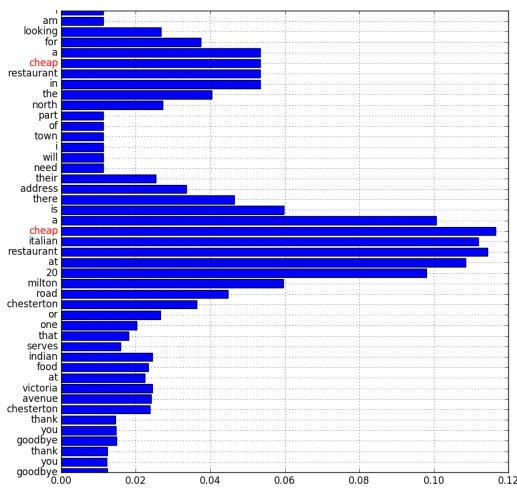

q=pricerange, a=cheap

Figure 2: Narrator output distribution after 100 rounds over a Cambridge dialog.

**Adaptive adversarial `GMemN2N` improves the accuracy** of the model on the Cambridge task by 2.3 points for a model with with 6 hops. The best performance on the TripAdvisor dataset was achieved by the adversarial GMemN2N with 4 hops. It improves the accuracy by 1.5 points.

The uniform protocol **improves the stability** of the performances compare to a standard reader but we went further with the adversarial protocol which improve both the overall accuracy and the stability of the performances. It is not clear for this task that the number of hops, between 4 and 6, has an influence on the general behaviour but we achieve the best performance with our adversarial protocol and a reader with 6 hops.

**All the average values of the models trained with the adversarial protocol are higher than the others**, even for the 5 hops model which doesn't achieve a very good max performance during the 10 replications we have run.

Performances on the CBT dataset are displayed in table 3. Because of the size of this dataset, we didn't repeat the training 10 times but only once. Results of the uniform training seem similar to the performances of the standard reader in this case but the accuracy of the models trained with our adversarial protocol remain higher than others.

## 5 VISUALIZATIONS AND ANALYSIS

### 5.1 NARRATOR PREDICTIONS

In order to better understand the narrator learnings from the reader behaviour during the adversarial protocol, Figure 2 depicts the rewards that the narrator expects for each word of a document after several rounds of the game. Given a tuple $(d, q)$ where $d$ is a clear document and $q$ a query and assuming the document contains $k$ words, we generate $k$ corrupted documents where one word is obfuscated in each of them. We then feed the narrator with these corrupted data and report the results. *y-axis* represents the document and *x-axis* the expected reward from the reader if the narrator decides to generate a corrupted document by obfuscating this word. In red, the words of the documents that correspond to the answer of the question are highlighted.

The narrator tends to obfuscate some important keywords of the dialogs. Furthermore, the narrator is not pointing on a single word but it points on a word and on its neighborhood. This might be a

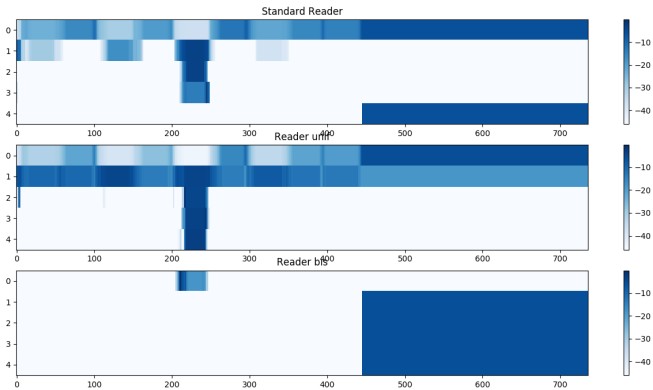

Figure 3: Reader attentions

consequence of the encoding which is not only a representation of a word but a representation of a word in its context.

## 5.2 STANDARD VS ADVERSARIAL READER ATTENTION

Figure 3 depicts the attention values, presented by hops, over a document from the Cambridge dataset. The document was choosen as only the adversarial protocol answers correctly to the question. It displays attention distributions for a reader trained with the three different protocols: [top] standard, [middle] uniform, [bottom] adversarial. The overall aspect of the first two readers are comparable. The readers quickly focus on what we assume to be an important span of text. After two hops the readers start *looking* at the same position in the document. On the contrary, the reader trained with the adversarial protocol seems to have a very different behavior regarding the attention mechanism. It captures the important part of the sentence directly at the first hop and uses the 4 remaining hops to focus more largely on the end of the document. We might interpret this as a consequence of the obfuscation protocol that forces the reader to look on different parts of the sentence instead of focusing on one precise point during the learning process.

## 6 RELATED WORK

### 6.1 END-TO-END MACHINE READING

The task of end-to-end machine reading consist in learning to select an answer to question given a passage of text in supervised manner. One of the popular formal setting of the problem, the cloze-style QA task, involves tuples of the form $(d, q, a, C)$, where $d$ is a document (context), $q$ is a query over the contents of $d$, in which a phrase is replaced with a placeholder, and a is the answer to $q$, which comes from a set of candidates $C$. In this work we consider datasets where each candidate $c \in C$ has at least one token which also appears in the document. The task can then be described as: given a document-query pair $(d, q)$, find $a \in C$ which answers $q$. Below we provide an overview of representative neural network architectures which have been applied to this problem.

LSTMs with Attention: Several architectures introduced in Hermann et al. (2015) employ LSTM units to compute a combined document-query representation $g(d, q)$, which is used to rank the candidate answers. These include the DeepLSTM Reader which performs a single forward pass through the concatenated (document, query) pair to obtain $g(d, q)$; the Attentive Reader which first computes a document vector $d(q)$ by a weighted aggregation of words according to attentions based on $q$, and then combines $d(q)$ and $q$ to obtain their joint representation $g(d(q), q)$; and the Impatient Reader where the document representation is built incrementally. The architecture of the Attentive Reader has been simplified recently in Stanford Attentive Reader, where shallower recurrent units were used with a bilinear form for the query-document attention (Chen et al., 2016).

Attention Sum: The Attention-Sum (AS) Reader (Kadlec et al., 2016) uses two bidirectional GRU networks to encode both $d$ and $q$ into vectors. A probability distribution over the entities in $d$ is obtained by computing dot products between $q$ and the entity embeddings and taking a softmax. Then, an aggregation scheme named pointer-sum attention is further applied to sum the probabilities of the same entity, so that frequent entities the document will be favored compared to rare ones. Building on the AS Reader, the Attention-over-Attention (AoA) Reader (Cui et al., 2016) introduces a two-way attention mechanism where the query and the document are mutually attentive to each other.

Multi-hop Architectures: Memory Networks (MemNets) were proposed in Weston et al. (2014), where each sentence in the document is encoded to a memory by aggregating nearby words. Attention over the memory slots given the query is used to compute an overall memory and to renew the query representation over multiple iterations, allowing certain types of reasoning over the salient facts in the memory and the query. Neural Semantic Encoders (NSE) Munkhdalai & Yu (2016) extended MemNets by introducing a write operation which can evolve the memory over time during the course of reading. Iterative reasoning has been found effective in several more recent models, including the Iterative Attentive Reader Sordoni et al. (2016) and ReasoNet Shen et al. (2016). The latter allows dynamic reasoning steps and is trained with reinforcement learning.

Other related works, included EpiReader (Trischler et al., 2016b), consist of two networks, where one proposes a small set of candidate answers, and the other reranks the proposed candidates conditioned on the query and the context; Bi-Directional Attention Flow network (BiDAF) (Seo et al., 2016) adopts a multi-stage hierarchical architecture along with a flow-based attention mechanism.

## 6.2 Adversarial Learning and Self-Play

The main principle of self-play consist in defining a learning task where two, possible antagonist behaviours, will be learnt jointly by competing from one against the another. In the context of two-player zero-sum games, such setting falls quite naturally. Two models of the same nature compete regarding the rules of the considered game and learn from their sucessive performances.

A majority of prior work has focused on learning from self-play data using temporal-difference learning in backgammon (Tesauro, 1995), chess (Mannen, 2003), or using linear regression in Othello (van der Ree & Wiering, 2013) and more recently Go (Silver et al., 2016). In the general context of board games, the main advantage of self-play as a method of training neural network controllers lies in the fact that every position will be the result of a game position from an actual board, rather than being contrived positions that may fail to teach the network about probabilities or prevent the network from properly generalizing from the results. In other word, self-play contributes to exhibit challenging configurations to overcome as a controller. In such setting, the network has the advantage of having seen over several million different board positions, which would have been hardly feasible in a network trained through a crafted set of training data.

In the domain of reading, it has been recently observed that the tasks of answering to a question given a passage of text and predicting the question regarding a text passage are interesting tasks to model jointly. So, several papers have recently proposed to use the question generation as a regularization task to improve the passage encoding model of a neural reader ((Yuan et al., 2017), (Wang et al., 2017)). In this paper, we claim these two tasks are indeed complementary but we think adversarial training of the nature used in two player games will lead to the same advantages than those observed previously. As generating the question given a passage and a question is hard we inspired ourself from the recent work proposed in (Guo et al., 2017) and define a narrator network as complementary task to the reader learning one. Such narrator have the task of finding the most meaningfull spans of text to obfuscate in a give passage and given a question in order to minimize the probability of successfull answering of the reader.

## 6.3 Adaptive dropout

Recent deep neural networks are composed of a lot of parameters and tend to easily overfit the training set. One of the main idea which has been developed to prevent this overfitting is to randomly drop units from the network during the training session (Srivastava et al., 2014). Such approach results to combine many different neural networks to make a prediction. In the same idea of avoiding

to overfit the training data, training a model on a dataset which contains corrupted data is something usefull which has been studying in Maaten et al. (2013). They have developed different ways to corrupt a document, for example by adding noise into the input features and our work refers to what they call the *blankout corruption* which consist of randomly delete features into the input documents (texts or images in this case) with probability $q$.

## 7 Conclusion and Future Work

In this paper, we propose an adversarial protocol to train coupled deep memory networks for the task of machine comprehension. On all reported experiments, the models trained with this novel protocol outperform the equivalent models trained using a standard supervised protocol. Moreover our adversarial protocol seems to reduce the variance of the models performances. In future work, we plan to continue studying this novel protocol using an active question answering task. Moreover, we currently investigate an adaptation of such protocol to Visual Question Answering.

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
