# OpenReview forum: "Adversarial reading networks for machine comprehension"
_ICLR.cc/2018/Conference — Reject_

### Official Review · AnonReviewer3 · 2017-11-16
**Interesting idea, unconvincing results**

**Rating:** 4
**Confidence:** 5

**Review:**

The paper aims to improve the accuracy of reading model on question answering dataset by playing against an adversarial agent (which is called narrator by the authors) that "obfuscates" the document, i.e. changing words in the document. The authors mention that word dropout can be considered as its special case which randomly drops words without any prior. Then the authors claim that smartly choosing the words to drop can make a stronger adversarial agent, which in turn would improve the performance of the reader as well. Hence the adversarial agent is trained and is architecturally similar to the reader but just has a different last layer, which predicts the word that would make the reader fail if the word is obfuscated.

I think the idea is interesting and novel. While there have been numerous GAN-like approaches for language understanding, very few, if any, have shown worthy results. So if this works, it could be an impactful achievement.

However, I am concerned with the experimental results.

First, CBT: NE and CN numbers are too low. Even a pure LSTM achieves (no attention, no memory) 44% and 45%, respectively (Yu et al., 2017). These are 9% and 6% higher than the reported numbers for adversarial GMemN2N. So it is very difficult to determine if the model is appropriate for the dataset in the first place, and whether the gain from the non-adversarial setting is due to the adversarial setup or not.

Second, Cambridge dialogs: the dataset's metric is not accuracy-based (while the paper reports accuracy), so I assume some preprocessing and altering have been done on the dataset. So there is no baseline to compare. Though I understand that the point of the paper is the improvement via the adversarial setting, it is hard to gauge how good the numbers are.

Third, TripAdvisor: the dataset paper by Wang et al. (2010) is not evaluated on accuracy (rather on ranking, etc.). Did you also make changes to the dataset? Again, this makes the paper less strong because there is no baseline to compare.

In short, the only comparable dataset is CBT, which has too low accuracy compared to a very simple baseline.
In order to improve the paper, I recommend the authors to evaluate on more common datasets and/or use more appropriate reading models.

---

Typos:
page 1 first para: "One the first hand" -> "On the first hand"
page 1 first para: "minimize to probability" -> "minimize the probability"
page 3 first para: "compensate" -> "compensated"
page 3 last para: "softmaxis" -> "softmax is"
page 4 sec 2.4: "similar to the reader" -> "similarly to the reader"
page 4 sec 2.4: "unknow" -> "unknown"
page 4 sec 3 first para: missing reference at "a given dialog"
page 5 first para: "Concretly" -> "Concretely"
Table 1: "GMenN2N" -> "GMemN2N"
Table 1: what is difference between "mean" and "average"?
page 8 last para: missing reference at "Iterative Attentive Reader"
page 9 sec 6.2 last para: several citations missing, e.g. which paper is by "Tesauro"?


[Yu et al. 2017] Adams Wei Yu, Hongrae Kim, and Quoc V. Le. Learning to Skim Text. ACL 2017

---

### Official Review · AnonReviewer1 · 2017-11-27
**Paper needs significant revision**

**Rating:** 5
**Confidence:** 5

**Review:**

Summary:

This paper proposes an adversarial learning framework for machine comprehension task. Specifically, authors consider a reader network which learns to answer the question by reading the passage and a narrator network which learns to obfuscate the passage so that the reader can fail in its task. Authors report results in 3 different reading comprehension datasets and the proposed learning framework results in improving the performance of GMemN2N.


My Comments:

This paper is a direct application of adversarial learning to the task of reading comprehension. It is a reasonable idea and authors indeed show that it works.

1. The paper needs a lot of editing. Please check the minor comments.

2. Why is the adversary called narrator network? It is bit confusing because the job of that network is to obfuscate the passage.

3. Why do you motivate the learning method using self-play? This is just using the idea of adversarial learning (like GAN) and it is not related to self-play.

4. In section 2, first paragraph, authors mention that the narrator prevents catastrophic forgetting. How is this happening? Can you elaborate more?

5. The learning framework is not explained in a precise way. What do you mean by re-initializing and retraining the narrator? Isn’t it costly to reinitialize the network and retrain it for every turn? How many such epochs are done? You say that test set also contains obfuscated documents. Is it only for the validation set? Can you please explain if you use obfuscation when you report the final test performance too? It would be more clear if you can provide a complete pseudo-code of the learning procedure.

6. How does the narrator choose which word to obfuscate? Do you run the narrator model with all possible obfuscations and pick the best choice?

7. Why don’t you treat number of hops as a hyper-parameter and choose it based on validation set? I would like to see the results in Table 1 where you choose number of hops for each of the three models based on validation set.

8. In figure 2, how are rounds constructed? Does the model sees the same document again and again for 100 times or each time it sees a random document and you sample documents with replacement? This will be clear if you provide the pseudo-code for learning.

9. I do not understand author's’ justification for figure-3. Is it the case that the model learns to attend to last sentences for all the questions? Or where it attends varies across examples?

10. Are you willing to release the code for reproducing the results?

Minor comments:

Page 1, “exploit his own decision” should be “exploit its own decision”
In page 2, section 2.1, sentence starting with “Indeed, a too low percentage …” needs to be fixed.
Page 3, “forgetting is compensate” should be “forgetting is compensated”.
Page 4, “for one sentences” needs to be fixed.
Page 4, “unknow” should be “unknown”.
Page 4, “??” needs to be fixed.
Page 5, “for the two first datasets” needs to be fixed.
Table 1, “GMenN2N” should be “GMemN2N”. In caption, is it mean accuracy or maximum accuracy?
Page 6, “dataset was achieves” needs to be fixed.
Page 7, “document by obfuscated this word” needs to be fixed.
Page 7, “overall aspect of the two first readers” needs to be fixed.
Page 8, last para, references needs to be fixed.
Page 9, first sentence, please check grammar.
Section 6.2, last sentence is irrelevant.

---

### Official Review · AnonReviewer2 · 2017-11-29
**The root idea is interesting but the paper has significant issues.**

**Rating:** 5
**Confidence:** 4

**Review:**

The main idea of this paper is to automate the construction of adversarial reading comprehension problems in the spirit of Jia and Liang, EMNLP 2017.  In that work a "distractor sentence" is manually added to a passage to superficially, but not logically, support an incorrect answer.  It was shown that these distractor sentences largely fool existing reading comprehension systems although they do not fool human readers.

This paper replaces the manual addition of a distractor sentence with a single word replacement where a "narrator" is trained adversarially to select a replacement to fool the question answering system.  This idea seems interesting but very difficult to evaluate.  An adversarial word replacement my in fact destroy the factual information needed to answer the question and there is no control for this.  The performance of the question answering system in the presence of this adversarial narrator is of unclear significance and the empirical results in the paper are very difficult to interpret.  No comparisons with previous work are given (and perhaps cannot be given).

A better model would be the addition of a distractor sentence as this preserves the information in the original passage.  A language model could probably be used to generate a compelling distractor.  But we want that the corrupted passage has the same correct answer as the uncorrupted passage and this difficult to guarantee.  A trained "narrator" could learn to actually change the correct answer.

---

### Decision · Program_Chairs · 2018-01-29
**ICLR 2018 Conference Acceptance Decision**

**Decision:**

Reject

**Comment:**

The paper presents an adversarial learning framework for reading comprehension.  Although the idea is interesting and presents an approach that ideally would make reading comprehension approaches more robust, the results are not substantially solid (see reviewer 3's comments) compared to other baselines to warrant acceptance.  Comments from reviewer 2 are also noteworthy where they mention that adversarial perturbations to a context around an answer can alter the facts in the context, thus destroying the actual information present there, and the rebuttal does not seem to satisfy the concern.  Addressing these issues will strengthen the paper for a potential future venue.